# The Strategies Used by Animal Viruses to Antagonize Host Antiviral Innate Immunity: New Clues for Developing Live Attenuated Vaccines (LAVs)

**DOI:** 10.3390/vaccines13010046

**Published:** 2025-01-08

**Authors:** Na Chen, Baoge Zhang

**Affiliations:** 1MOE Joint International Research Laboratory of Animal Health and Food Safety, Engineering Laboratory of Animal Immunity of Jiangsu Province, College of Veterinary Medicine, Nanjing Agricultural University, Nanjing 210095, China; chennafz@stu.njau.edu.cn; 2College of Veterinary Medicine, Nanjing Agricultural University, Nanjing 210095, China

**Keywords:** animal viruses, antagonism, host antiviral innate immunity, interferon production, interferon responses, pathogenicity, LAVs

## Abstract

As an essential type of vaccine, live attenuated vaccines (LAVs) play a crucial role in animal disease prevention and control. Nevertheless, developing LAVs faces the challenge of balancing safety and efficacy. Understanding the mechanisms animal viruses use to antagonize host antiviral innate immunity may help to precisely regulate vaccine strains and maintain strong immunogenicity while reducing their pathogenicity. It may improve the safety and efficacy of LAVs, as well as provide a more reliable means for the prevention and control of infectious livestock diseases. Therefore, exploring viral antagonistic mechanisms is a significant clue for developing LAVs, which helps to explore more viral virulence factors (as new vaccine targets) and provides a vital theoretical basis and technical support for vaccine development. Among animal viruses, ASFV, PRRSV, PRV, CSFV, FMDV, PCV, PPV, and AIV are some typical representatives. It is crucial to conduct in-depth research and summarize the antagonistic strategies of these typical animal viruses. Studies have indicated that animal viruses may antagonize the antiviral innate immunity by directly or indirectly blocking the antiviral signaling pathways. In addition, viruses also do this by antagonizing host restriction factors targeting the viral replication cycle. Beyond that, viruses may antagonize via regulating apoptosis, metabolic pathways, and stress granule formation. A summary of viral antagonistic mechanisms might provide a new theoretical basis for understanding the pathogenic mechanism of animal viruses and developing LAVs based on antagonistic mechanisms and viral virulence factors.

## 1. Introduction

Infectious livestock diseases have a crucial impact on the development of animal husbandry and public health safety. In the field of animal husbandry, the outbreak of diseases may lead to the death of a large number of animals (including pigs, poultry, cows, and sheep), a decrease in production performance, and ultimately huge economic losses to farmers. In addition, infectious livestock diseases may also spread to humans, triggering a public health crisis. For example, viruses such as avian influenza not only threaten poultry farming but also pose a potential threat to human health [1,2]. Therefore, effective prevention and control of infectious livestock diseases is a key task in ensuring the sustainable development of animal husbandry and public health safety. With the continuous development of global trade, the spread rate and scope of infectious livestock diseases are also constantly expanding, further increasing the difficulty of prevention and control. Governments and relevant institutions worldwide are increasing their investment in animal disease prevention and control to ensure the stable development of animal husbandry and public health safety.

LAVs play an essential role in animal disease prevention and control. They stimulate the body to produce a comprehensive and long-lasting immune response by simulating the process of natural infection. Compared with other types of vaccines, LAVs have many advantages. Firstly, they can induce cellular and humoral immunity, providing broader protection. Secondly, LAVs usually require fewer doses and less immunization, reducing the costs and difficulty of vaccination [3]. In history, LAVs have achieved significant results in controlling many infectious livestock diseases and contributed to the stable development of animal husbandry. For example, the successful development and application of the cowpox vaccine completely eradicated smallpox and brought tremendous benefits to human health [4]. In animal disease prevention and control, LAVs have also played a crucial role, such as the live attenuated classical swine fever vaccine and the live attenuated Newcastle disease virus vaccine, which have made positive contributions to controlling the spread of corresponding diseases [5,6]. It is worth mentioning that the virulence of viruses was attenuated classically by passing many times through different hosts or tissues or at various temperatures. Different host- or tissue-adapted viruses show the attenuated phenotype for the natural host infection. Different temperature-adapted viruses show the attenuated phenotype during the natural infection, thereby showing the ts (temperature sensitive)-phenotype. This is how classical LAVs have developed. For example, the human live vaccines, measles, mumps, and rubella vaccines have been established by adapting other hosts (i.e., chick embryonic fibroblasts, rabbit kidney) and have been successfully used worldwide [7,8]. The live influenza vaccine (FluMist) with ts-phenotype can grow only in the upper respiratory tissue with a low temperature (but not in the lower respiratory tissues with an average body temperature) and induce protective immunity safely [9,10]. Recently, some researchers have summarized how viruses are attenuated during the adaptation process from the perspective of viral replication cycles, including the viral entry step, modulating viral RNA/DNA synthesis, translation stage, and restricting cell-to-cell virus spread [11].

It has been reported that the viral virulence factors involved in antagonizing host innate immunity (including antagonizing interferon production and interferon responses) are new targets and clues for LAVs [3,12,13,14,15,16,17,18]. The modification or deletion of these virulence factors is a promising strategy for developing novel LAVs. Recently, Tang et al. systematically summarized the future research directions of LAVs, which also emphasized this strategy [11]. Researchers attempted to use reverse genetics technology to construct mutant viruses with a phenotype lacking interferon antagonism. Subsequently, interferon-antagonism-negative viruses were shown to induce better immune responses [12]. For example, the PRRSV nsp1β is an effective interferon antagonist that can inhibit interferon production and interferon responses. Mutant PRRSVs were generated to eliminate the interferon-antagonism function from nsp1β [12]. Pigs infected with nsp1β-mutant PRRSVs exhibited multiple changes, including higher levels of interferon-α, ISGs, interferon-γ, and NK cell function, as well as shorter duration and lower titers of viremia and strong PRRSV-specific antibody responses [19,20]. In addition, the interferon-antagonism-negative PRRSV mutants are clinically attenuated, as evidenced by the fact that the neutralizing antibody titers were higher compared to those of control pigs [20]. These studies indicate the role of type I interferon in triggering adaptive immune responses in pigs, and provide evidence that interferon-antagonism-negative PRRSV may be developed as a new vaccine candidate. Removing interferon antagonism from the virus represents a reasonable strategy for developing next-generation vaccines. Studies have shown that FMDV L^pro^ antagonizes interferon production and interferon responses through multiple pathways [21,22,23]. Therefore, L^pro^ is the target for the design of a live-attenuated FMDV vaccine. The experimental results indicate that after removing the interferon-antagonism function from L^pro^, interferon-antagonism-negative FMDV elicits higher levels of interferons and ISGs, accompanied by a potent neutralizing antibody response in vaccinated pigs [12,24,25].

Nonetheless, developing LAVs has never been smooth sailing. Currently, the balance between safety and efficacy is a key challenge in developing LAVs. On the one hand, to ensure the safety of vaccines, it is necessary to attenuate the virus, but excessive attenuation may lead to insufficient immunogenicity, which cannot effectively stimulate host immune responses. On the other hand, if the attenuation is not sufficient, the vaccine may retain high virulence, which poses a risk of causing disease [3,17]. Several studies have shown that the ability of viruses to antagonize host antiviral innate immunity is often closely related to virus virulence. Generally speaking, antagonistic ability is positively correlated with viral virulence [17]. Therefore, grasping this balance requires a deep understanding of viral antagonistic mechanisms, which helps to relatively precisely regulate vaccine strains and maintain good immunogenicity while reducing their pathogenicity. This can not only improve the safety and efficacy of LAVs but also provide a more reliable means for the prevention and control of infectious livestock diseases. For instance, influenza A virus (IAV) NS1 may antagonize multiple signal transduction processes at multiple levels, such as transcription and translation, suggesting that NS1 is an important virulence factor of IAV [26]. Given this, researchers use reverse genetics technology to truncate or delete the NS1 gene, thereby reducing the ability of IAV antagonism. Immunization with the rescued candidate vaccine strain could induce strong humoral and cellular responses and produce immune protection against wild-type IAV challenges [27].

In this review, we attempt to deeply explore the strategies of animal viruses to antagonize host antiviral innate immunity, as well as the potential applications of these strategies in developing LAVs. Through a systematic review and analysis of relevant research results, we hope to provide more viral virulence factors (as new vaccine targets) and new clues for developing LAVs. In light of the background of the severe situation of animal disease prevention and control, it is of great practical significance to deeply explore viral antagonistic mechanisms. We believe that through continuous in-depth research and exploration, safer and more effective LAVs can be developed, providing a more reliable means for the prevention and control of infectious livestock diseases.

## 2. Antiviral Innate Immune Signaling Pathways

Viral infections could trigger both innate and adaptive immune reactions. Antiviral innate immune signaling pathways are of crucial significance in innate immune responses. Pattern recognition receptors (PRRs) may detect viral nucleic acids and activate transcription factors like nuclear factor κappa-light-chain-enhancer of activated B cells (NF-κB), interferon regulatory factor 3 (IRF3), and IRF7. Throughout this process, vital signaling proteins, such as mitochondrial antiviral signaling (MAVS) and IκB Kinase α (IKKα)/IKKβ/NF-κB essential modulator (NEMO) complexes, play a role in signal transduction [26]. Once activated, these transcription factors induce IFNα/β production. Immediately thereafter, IFNα/β binds to interferon-α/β receptor (IFNAR) and activates the Janus kinase/signal transducers and activators of transcription (JAK/STAT) signaling pathways, thereby initiating IFN-stimulated gene (ISG) synthesis and inhibiting viral replication [28].

There has been an ongoing battle between viruses and hosts. Once they recognize viral nucleic acid, host PRRs rapidly trigger antiviral innate immune signaling pathways to initiate the synthesis of ISGs, thereby inhibiting viral replication. Nonetheless, animal viruses have evolved various antagonistic mechanisms to directly or indirectly block the signaling pathways. Consequently, it is a significant clue for developing LAVs to explore these potential mechanisms.

## 3. The Antagonistic Mechanisms for African Swine Fever Virus (ASFV)

### 3.1. Brief Introduction to ASFV

ASFV is the causative agent of African swine fever (ASF) and a double-stranded DNA virus belonging to the Asfarviridae family. The ASFV genome has a size range of 170–193 kbp and encodes 150–190 open reading frames (ORFs) [29]. After being infected with ASFV, pigs usually experience constitutional symptoms such as high fever, skin cyanosis, bleeding, and dyspnea. The disease has a rapid onset and a mortality rate of up to 100%, posing a great threat to pig herds. However, there is currently no safe vaccine or treatment available [30]. In addition, little is known about the viral proteins and mechanisms responsible for antagonizing the host antiviral innate immunity, which seriously hinders the development of vaccines. The research on LAVs against ASFV has always been of great concern. Currently, researchers are committed to developing LAVs with reduced virulence but still able to stimulate effective immune responses through in-depth research and modification of viral genes. Nonetheless, there are many challenges in the development process of LAVs, such as the balance between safety and efficacy [29,31,32,33]. A deeper understanding of the antagonistic mechanisms of ASFV will help to better understand the pathogenic process of ASFV, provide new clues for developing and optimizing LAVs based on viral antagonistic mechanisms and virulence factors, and also provide stronger support for the prevention and control of African swine fever.

### 3.2. Blocking Antiviral Innate Immune Signaling Pathways Directly

As PRRs, cyclic GMP-AMP synthase (cGAS) and stimulation of interferon genes (STING) are significant for innate immune responses. Nonetheless, ASFV has evolved several strategies to block PRRs signaling. For example, ASFV QP383R may interact with cGAS to facilitate cGAS palmitoylation, thereby inhibiting the synthesis of cGAMPs and type I interferon-mediated antiviral immune responses [34]. Several studies have shown that ASFV B175L, E184L, and p17 could interact with STING to disrupt STING-TBK1-IRF3 complex formation, resulting in inhibition of IRF3 phosphorylation and interferon production [35,36,37]. In addition, ASFV L83L, MGF360-13L, and MGF505-6R target STING for autophagic degradation [38,39,40]. By contrast, ASFV MGF505-11R interacts with STING and degrades it through the lysosomal, ubiquitin-proteasome, and autophagy pathways [41].

Remarkably, STING recruits TBK1 and IKKε, resulting in the phosphorylation and nuclear translocation of IRF3. In addition to inducing IRF3 phosphorylation, the activated STING interacts with the IKKα/IKKβ/NEMO complexes, which then activate NF-κB signaling. Multiple studies have shown that ASFV could block this signal transduction through different mechanisms. Specifically, ASFV A137R targets TBK1 for lysosomal degradation, while MGF-110-9L targets it for autophagic degradation [42,43]. In addition, ASFV S273R interacts with IKKε and disturbs the interaction between IKKε and STING to negatively regulate the cGAS-STING signaling pathway [44]. Furthermore, ASFV E120R, MGF360-4L, and S273R reduce interferon-β expression by inhibiting IRF3 phosphorylation [45,46,47]. By contrast, ASFV M1249L and MGF360-14L target IRF3 and induce its degradation through the lysosomal and ubiquitin-proteasome pathways, respectively [48,49]. Moreover, E301R, DP96R, and MGF505-7R may inhibit type I interferon production by interacting with IRF3 and impairing its nuclear translocation [50,51,52]. Interestingly, ASFV D345L interacts with IKKα and IKKβ to negatively regulate NF-κB signaling by inhibiting IKK kinase activity [53]. Beyond that, ASFV F317L interacts with IKKβ and inhibits its phosphorylation, resulting in the inhibition of NF-κB activation [54].

Upon activation, the transcription factors IRF3 and NF-κB bind to the interferon promoter to modulate interferon expression and initiate interferon responses. The binding of interferons and interferon receptors activates the JAK-STAT signaling pathway. However, ASFV could block it through multiple mechanisms. Specifically, ASFV B318L, CD2v, H240R, and K205R interact with IFNAR1 and IFNAR2 to inhibit the recruitment of tyrosine kinase 2 (TYK2) and JAK1 by IFNAR1 and IFNAR2, which leads to reduced phosphorylation and translocation of STAT1 and STAT2 [55,56,57,58]. Ubiquitylation can be divided into two types: the degradative type (such as K48-linked polyubiquitination) and the non-degradative type (such as K63-linked polyubiquitination). ASFV MGF-360-10L is reported to mediate the K48-linked ubiquitination of JAK1 and its degradation through ubiquitin-proteasome pathways by recruiting the E3 ubiquitin ligase HERC5 [59]. Furthermore, ASFV MGF360-9L interacts with STAT1 and mediates its degradation through the apoptosis pathway [60]. In addition, ASFV B475L negatively regulates interferon signaling through interacting with STAT2 and ultimately preventing the nuclear translocation of STAT1 and STAT2 [61]. By contrast, ASFV MGF360-9L, I215L, and S273R target STAT2 for degradation through ubiquitin-proteasome pathways [60,62,63]. The IFN-stimulated gene factor 3 (ISGF3) complex consists of STAT1, STAT2, and IRF9. Interestingly, ASFV MGF505-7R targets IRF9 to inhibit ISGF3 heterotrimer formation and the nuclear translocation of ISGF3, while I215L targets IRF9 for degradation via the autophagy-lysosome pathway [64,65].

### 3.3. Blocking Antiviral Innate Immune Signaling Pathways Indirectly by Hijacking Proviral Host Factors and Antagonizing Host Restriction Factors

Host factors associated with viral replication can be categorized into two types: proviral host factors and host restriction factors, which directly or indirectly regulate viral replication. Unc-51, like autophagy-activating kinase 1 (ULK1) and Riplet, function as proviral host factors and host restriction factors for ASFV, respectively. Studies have shown that ASFV could block antiviral signaling pathways indirectly by hijacking proviral host factors and antagonizing host restriction factors. Mechanistically, ASFV MGF-505-7R interacts with ULK1 and enhances its expression, which eventually leads to the degradation of STING and thus inhibits the cGAS-STING pathway [66]. In addition, ASFV I267L interacts with Riplet to impair Riplet-mediated RIG-I activation and interferon-β expression [67]. These studies enrich our understanding of the antagonistic mechanisms of ASFV.

### 3.4. Antagonizing Host Restriction Factors Targeting the Viral Replication Cycle

Recent studies have shown that host restriction factor forkhead box J1 (FoxJ1) inhibits ASFV replication by targeting a late step in the viral replication cycle. Mechanistically, FoxJ1 degrades ASFV MGF505-2R and E165R via the autophagy pathway. Further studies demonstrated that ASFV S273R suppresses the expression of FoxJ1 in a dose-dependent fashion. Taken together, ASFV S273R impairs the FoxJ1-mediated antiviral effect by decreasing its expression [68]. Remarkably, host restriction factor FoxJ1 inhibits viral replication by targeting and modulating the viral replication cycle rather than the signaling pathways, which may expand our understanding of the mechanism of ASFV antagonism.

### 3.5. Regulation of Metabolic Pathways, Stress Granule Formation, and Apoptosis

In addition to the mechanisms described above, ASFV also antagonizes innate immunity by regulating metabolic pathways, the formation of stress granules, and apoptosis. Specifically, ASFV infection induces lactate production, which promotes ASFV replication by reducing interferon-β induction [69]. In response to environmental stimuli, such as viral infections, eukaryotic cells could induce the formation of stress granules (SGs) to resist viruses. Interestingly, ASFV S273R antagonizes stress granule formation and promotes viral replication by cleaving the nucleating protein G3BP1 [70]. By contrast, ASFV may maintain de novo protein synthesis and inhibit stress granule formation via dephosphorylating eukaryotic initiation factor 2α (eIF2α) [71].

According to reports, the JAK2-STAT3 pathway can be activated by cytokines such as interleukin and interferon. It participates in regulating various biological functions such as cell apoptosis, inflammation, proliferation, and differentiation. Studies have shown that ASFV infection triggers the activation of the JAK2-STAT3 pathway in the early phases, while it activates apoptosis in the late phases. The JAK2-STAT3 pathway is crucial for ASFV replication, whereas apoptosis inhibits viral replication. Interestingly, ASFV CD2v could interact with colony stimulating factor 2 receptor alpha (CSF2RA) to activate the JAK2-STAT3 pathway and inhibit apoptosis, ultimately maintaining the survival of infected cells and facilitating viral replication [72]. These studies extend the research direction on antagonistic mechanisms of ASFV.

## 4. The Antagonistic Mechanisms of Porcine Reproductive and Respiratory Syndrome Virus (PRRSV)

### 4.1. Brief Introduction to PRRSV

PRRSV is the causative agent of porcine reproductive and respiratory syndrome (PRRS) and a single-stranded positive-sense RNA virus belonging to the Arteriviridae family. The PRRSV genome is approximately 15.4 kb and consists of at least 11 ORFs [73]. Once infected with PRRSV, pigs will experience a series of serious symptoms. In terms of reproduction, it manifests as reproductive disorders such as miscarriage and stillbirth. In terms of the respiratory system, symptoms such as dyspnea and coughing may occur, seriously affecting the health status of pigs. These symptoms not only cause great pain to infected pigs but also spread rapidly within the pig population, leading to a large number of pigs falling ill and posing extremely serious harm to the entire pig population [74]. Although some virulence factors of PRRSV that are involved in antagonizing interferon responses have been identified as targets for the developing LAVs, there are still some issues with their clinical application, such as possible virulence reversion [3,74]. In-depth exploration of the antagonistic mechanisms of PRRSV may provide key clues for further optimization of LAVs, laying the foundation for developing safer and more effective LAVs, and providing strong support for the prevention and control of PRRS.

### 4.2. Blocking Antiviral Innate Immune Signaling Pathways Directly

Studies have shown that PRRSV Nsp3 decreases the expression of melanoma differentiation-associated gene 5 (MDA5) through selective autophagy degradation, while Nsp11 reduces mRNA and protein levels of RIG-I and MAVS [75,76]. In addition, PRRSV Nsp4 suppresses the induction of interferon-β by cleaving MAVS in a proteasome- and caspase-independent manner [77]. Additionally, PRRSV Nsp4 interacts with IKKβ and then cleaves it, thereby inhibiting the activation of the NF-κB signaling pathway [78]. Furthermore, PRRSV Nsp4 suppresses the NF-κB signaling pathway by cleaving NEMO, which results in the downregulation of interferon-β expression [79].

Interestingly, PRRSV Nsp11 suppresses interferon signaling by interacting with STAT2 and inducing its degradation via the ubiquitin-proteasome pathways [80]. In addition, PRRSV Nsp11 could interact with the IRF-association domain of IRF9 and the formation of ISGF3, while Nsp1β may impair ISGF3 nuclear translocation [81,82]. ISG15 and tripartite motif protein 25 (TRIM25) belong to the ISGs. It is reported that PRRSV Nsp11 and Nsp1α could degrade them through autophagy-lysosome and proteasome systems, respectively [83,84].

### 4.3. Blocking Antiviral Innate Immune Signaling Pathways Indirectly by Hijacking Proviral Host Factors and Noncoding RNAs (ncRNAs)

It is reported that PRRSV Nsp1α may hijack proviral host factor ankyrin repeat- and SOCS box-containing 8 (ASB8) to promote the K48-linked ubiquitination and degradation of IKKβ through the ubiquitin-proteasome pathways. As a result, it causes remarkable inhibition of the phosphorylation of IκBα and p65, consequently inhibiting NF-κB activity [85].

In addition, although host restriction-factor heat shock protein 60 (HSP60) could interact with MAVS to positively modulate type I interferon responses, PRRSV infection induces miR-382-5p expression to negatively regulate HSP60 expression [86]. By contrast, miR-216 induced by PRRSV infection inhibits type I interferon production by targeting MAVS 3′UTR [87]. Several studies have shown that PRRSV infection also antagonizes antiviral innate immunity by inducing the expression of miR-541-3p, miR-30c, miR-376b-3p, and lncRNA-LOC103222771. Mechanistically, miR-541-3p targets the transcription factor IRF7 and reduces its expression to block interferon responses and promote PRRSV-2 replication [88]. In addition, miR-30c targets the 3′UTR region of IFNAR2 and JAK1 and negatively regulates their expression, whereas miR-376b-3p targets TRIM22 (as an ISG) [89,90,91]. Interestingly, claudin-4 is reported to inhibit PRRSV replication. Nevertheless, lncRNA-LOC103222771 induced by PRRSV infection could promote viral replication by downregulating host restriction factor claudin-4 and antagonizing its antiviral effect [92].

### 4.4. Blocking Antiviral Innate Immune Signaling Pathways Indirectly by Antagonizing Host Restriction Factors

Although a laboratory of genetics and physiology 2 (LGP2), TRIM25, DEAD-box helicase 10 (DDX10), and histone deacetylase 2 (HDAC2) are reported to be host restriction factors, PRRSV could antagonize them in different ways. Mechanistically, LGP2 facilitates the interaction between MDA5 and dsRNA to enhance antiviral responses. However, PRRSV Nsp1 and Nsp2 interact with LGP2 to weaken this antiviral effect [93]. In addition, PRRSV N suppresses TRIM25 expression and impairs the interaction between TRIM25 and RIG-I to inhibit the ubiquitination of RIG-I mediated by TRIM25, indirectly blocking antiviral signaling pathways [94].

A recent study has shown that DDX10 activates the type I interferon signaling pathways and induces ISGs expressions to inhibit PRRSV proliferation. Conversely, PRRSV E mediates the SQSTM1-dependent selective autophagic degradation of DDX10 by promoting its translocation from the nucleus to the cytoplasm [95]. By contrast, HDAC2 positively regulates interferon responses, including increasing the expression of ISG15, ISG54, and ISG56. Nonetheless, PRRSV Nsp11 mediates HDAC2 reduction in an endonuclease activity-dependent manner and antagonizes HDAC2-mediated antiviral responses [96].

### 4.5. Regulation of Metabolic Pathways

Interestingly, it is reported that PRRSV infection induces lactate production, which facilitates PRRSV replication by limiting interferon-β induction in different ways. Mechanistically, PRRSV-induced lactate upregulates cellular lactylation, which enhances the mRNA and protein levels of proviral host factor heat shock 70 kDa protein 6 (HSPA6). HSPA6 targets IKKε to impair IKKε recruitment by TNF receptor-associated factor 3 (TRAF3), thereby negatively regulating interferon-β production [97]. On the other hand, lactate reduces interferon-β expression and promotes viral replication by targeting MAVS and inhibiting MAVS-mediated antiviral responses [98].

## 5. The Antagonistic Mechanisms for Pseudorabies Virus (PRV)

### 5.1. Brief Introduction to PRV

PRV is the causative agent of pseudorabies (PR) and a double-stranded DNA virus belonging to the Herpesviridae family. The PRV genome is approximately 150 kb and encodes over 70 genes. PRV can infect various mammals, including pigs, sheep, and cows, leading to severe clinical symptoms and death. After being infected with PRV, pigs may experience neurological symptoms (such as ataxia and seizures) and respiratory symptoms, posing a serious threat to animal health [99]. Currently, LAVs are used worldwide to prevent PRV infection in pigs. However, an increasing amount of data suggests that these vaccines fail to provide complete protection against the new PRV variants [17]. A more in-depth analysis of the antagonistic mechanisms of PRV may provide new entry points for developing LAVs based on antagonistic mechanisms and provide a strong basis for developing more efficient LAVs.

### 5.2. Blocking Antiviral Innate Immune Signaling Pathways Directly

A recent study has shown that PRV UL21 inhibits the antiviral innate immune responses by triggering cGAS degradation. Mechanistically, PRV UL21 scaffolds the E3 ligase UBE3C to catalyze the K27-linked ubiquitination of cGAS at Lys384, which is degraded in the lysosome [100]. In addition, PRV UL38 interacts with STING and subsequently initiates the degradation of STING via selective autophagy, while US2 binds to STING and recruits TRIM21, thereby inducing STING K48-linked ubiquitination and degrading STING via the proteasome pathway [101,102].

Additionally, PRV US3 reduces interferon-β expression by targeting and inhibiting IRF3 phosphorylation [103]. By contrast, PRV UL13 and UL41 could target IRF3 and induce its degradation through the ubiquitin-proteasome and tollip-mediated autophagy pathways, respectively [104,105]. Several studies have shown that PRV may inhibit type I interferon responses by different mechanisms. Specifically, PRV UL50 induces lysosomal degradation of IFNAR1 and antagonizes the type I interferon responses [106]. Furthermore, PRV EP0 reduces IRF9 transcription and expression via its C-terminal region, while UL42 interacts with IFN-stimulated response elements (ISRE) and hinders ISGF3’s ability to bind to ISRE for efficient gene transcription [107,108].

### 5.3. Blocking Antiviral Innate Immune Signaling Pathways Indirectly by Hijacking Proviral Host Factors

Interestingly, PRV UL13 could also hijack proviral host factor RING finger protein 5 (RNF5) to suppress STING-mediated antiviral immunity. Mechanistically, PRV UL13 interacts with STING and recruits RNF5 to promote K27-/K29-linked ubiquitination and degrade STING [109]. Additionally, recent studies have shown that PRV may hijack proviral host factor TRIM26 to weaken the MAVS-mediated antiviral responses. Mechanistically, PRV infection induces TRIM26 expression, and TRIM26 promotes PRV replication through autophagic degradation of MAVS [110].

### 5.4. Blocking Antiviral Innate Immune Signaling Pathways Indirectly by Antagonizing Host Restriction Factors

Leucine-rich PPR motif-containing protein (LRPPRC), peroxiredoxin 1 (PRDX1), and Bcl-2-associated transcription factor 1 (Bclaf1) are host restriction factors for PRV. It is reported that PRV could block antiviral signaling pathways indirectly by antagonizing these host restriction factors. Specifically, LRPPRC interacts with STING and enhances NF-κB activation and interferon-β expression mediated by STING. Nonetheless, PRV UL16 could interact with LRPPRC to weaken the activation effect of LRPPRC on STING-mediated NF-κB signaling [111]. In addition, PRDX1 positively regulates interferon signaling via interactions with TBK1 and IKKε. However, PRV UL13 may mediate PRDX1 proteolysis in the proteasome degradation pathway via K48-linked ubiquitination [112]. Furthermore, PRV US3 is found to degrade Bclaf1 in a proteasome-dependent manner, ultimately inhibiting Bclaf1-mediated STAT1/STAT2 phosphorylation and the association of ISGF3 with DNA [113].

## 6. The Antagonistic Mechanisms for Classical Swine Fever Virus (CSFV)

### 6.1. Blocking Antiviral Innate Immune Signaling Pathways Directly

CSFV is the causative agent of classical swine fever (CSF) and a positive-strand RNA virus belonging to the Flaviviridae family. The CSFV genome is approximately 12 kb [114]. A deeper understanding of the antagonistic mechanisms of CSFV will lay a theoretical foundation for developing LAVs based on antagonistic mechanisms and viral virulence factors in the future. It has been demonstrated that TRAF6 interacts with CSFV NS3, thereby inhibiting viral replication via the NF-κB signaling pathway. However, TRAF6 was degraded during CSFV infection, which might contribute to persistent viral replication [115]. In addition, CSFV N^pro^ could target IRF3 and induce its degradation through ubiquitin-proteasome pathways [116]. It has been reported that CSFV N^pro^ interacts with IRF7 to reduce IRF7-dependent interferon-α induction in plasmacytoid dendritic cells (pDC). Further research found that the Zn-binding domain of N^pro^ is required for the interaction with IRF7 [117].

### 6.2. Blocking Antiviral Innate Immune Signaling Pathways Indirectly by Antagonizing Host Restriction Factors

Although host restriction factor hemoglobin subunit beta (HB) interacts with RIG-I to increase RIG-I expression and upregulate RIG-I-mediated type I interferon induction, CSFV C interacts with HB to inhibit the expression of HB and RIG-I, thereby antagonizing the HB-mediated antiviral effect [118]. Moreover, host restriction factor thioredoxin 2 (Trx2) is reported to inhibit CSFV replication via the NF-κB signaling pathway. Surprisingly, CSFV E2 could induce the downregulation of Trx2 to antagonize Trx2-mediated antiviral responses [119]. In addition, CSFV N^pro^ may enhance high mobility group box 1 (HMGB1) acetylation and its degradation by lysosomes to suppress the enhancement of interferon responses mediated by HMGB1 [120].

### 6.3. Regulation of Metabolic Pathways

Interestingly, CSFV may have the potential to antagonize antiviral innate immunity by regulating metabolic pathways. Specifically, CSFV inhibits serine metabolism-mediated antiviral innate immune responses by deacetylating modified phosphoglycerate dehydrogenase (PHGDH) [121]. In addition, CSFV infection induces indoleamine 2,3-dioxygenase 1 (IDO1) expression, thereby promoting tryptophan metabolism. IDO1 may negatively regulate the NF-κB signaling by mediating tryptophan metabolism, ultimately facilitating viral replication [122].

## 7. The Antagonistic Mechanisms for Foot and Mouth Disease Virus (FMDV)

### 7.1. Blocking Antiviral Innate Immune Signaling Pathways Directly

FMDV is the causative agent of foot and mouth disease (FMD) and a single-stranded positive-sense RNA virus belonging to the Picornaviridae family. The FMDV genome is approximately 8.5 kb [123]. Fortunately, people have begun to attempt to apply viral virulence factors that participate in antagonizing the host innate immunity to developing LAVs against FMDV, which has been briefly introduced above. Therefore, summarizing more viral antagonistic mechanisms and viral virulence factors can provide a theoretical basis for vaccine development [18,24,25]. A recent study has shown that FMDV 3C^pro^ and L^pro^ target cGAS for cleavage to dampen the cGAS/STING-dependent antiviral responses [21]. Moreover, the N-terminal 51 amino acids of FMDV 3A could interact with MDA5 to inhibit MDA5 expression by decreasing its mRNA levels, not at translational levels [124]. In addition, FMDV 3C^pro^ directly interacts with MDA5 and reduces its protein level through 3C^pro^ protease activity [125]. By contrast, FMDV L^pro^ negatively regulates MDA5-mediated antiviral responses by cleaving MDA5 at a conserved helicase motif [126]. Interestingly, FMDV 2B and 3A may antagonize RIG-I-mediated antiviral effects by inhibiting RIG-I expression [124,127]. In addition, FMDV 3B is reported to prevent the K63-linked ubiquitination of RIG-I, thereby inhibiting the formation of the RIG-I-MAVS complex and antiviral responses [128]. Furthermore, FMDV 3A could inhibit MAVS expression by disrupting its mRNA levels, ultimately reducing MAVS-mediated interferon-β induction [124]. Remarkably, FMDV VP3 targets the TM domain of MAVS to impede its mitochondrial localization, ultimately inhibiting the activation of interferon signaling [129].

Interestingly, FMDV 3C^pro^ specifically targets and cleaves NEMO, which impairs the ability of NEMO to activate downstream interferon production. Mutations specifically of 3C^pro^ abrogate NEMO cleavage and inhibit interferon induction [130]. By contrast, FMDV L^pro^ targets p65/RelA and induces its degradation, thus affecting NF-κB activity [22]. Additionally, FMDV L^pro^ antagonizes type I interferon signaling by targeting and cleaving STAT1 and STAT2, whereas 3C^pro^ antagonizes it by blocking STAT1/STAT2 nuclear translocation [23,131]. Moreover, FMDV 3C^pro^ also antagonizes host antiviral innate immunity by influencing the activity of ISGs such as PKR [132].

### 7.2. Blocking Antiviral Innate Immune Signaling Pathways Indirectly by Antagonizing Host Restriction Factors

Several studies have shown that LGP2, HDAC5, and HDAC8 are host restriction factors for FMDV. Interestingly, although LGP2 inhibits FMDV replication, FMDV 2B could induce a reduction in LGP2 and antagonize the LGP2-induced antiviral effect [133]. In addition, HDAC5 promotes IRF3 phosphorylation and IRF3-mediated antiviral innate immune responses during FMDV infection. HDAC8 increases the phosphorylation of TBK1 and IRF3, thereby promoting ISGs expression. To counteract their antiviral function, FMDV VP1 and VP3 could degrade HDAC5 and HDAC8 through the ubiquitin-proteasome and autophagy pathways, respectively [134,135].

### 7.3. Antagonizing Host Restriction Factors Targeting the Viral Replication Cycle

It has been reported that host restriction factors vacuolar protein sorting 28 (Vps28) and zinc finger protein 36 (ZFP36) may regulate different steps of the viral replication cycle. Specifically, Vps28 degrades viral structural proteins VP0, VP1, and VP3. ZFP36 targets the VP3 and VP4 for degradation through ubiquitin-proteasome pathways. Nevertheless, FMDV 3C^pro^ protease degrades Vps28 by utilizing autophagy and its protease activity, while 3C^pro^ degrades ZFP36 via its protease activity [136,137].

## 8. The Antagonistic Mechanisms for Other Animal Viruses

In addition to the five animal viruses mentioned above, porcine circovirus (PCV), porcine parvovirus (PPV), and avian influenza virus (AIV) have also had a significant impact on the livestock industry, causing significant economic losses that cannot be ignored [138,139,140]. It is particularly important to conduct in-depth research on their antagonistic mechanisms, which will provide new ideas for developing LAVs based on viral antagonistic mechanisms and virulence factors. It has been reported that PCV2 Cap could induce dnaJ homolog subfamily C member 7 (DNAJC7) expression to inhibit PKR activation, while PCV3 Cap may inhibit ISGF3 binding to the ISRE promoter [138,141]. In addition, PPV NS2 inhibits dsRNA-induced interferon-β promoter activation [139]. Additionally, several studies indicated that AIV could block antiviral signaling pathways directly or indirectly through different mechanisms. Specifically, AIV PB1 inhibits innate immune responses by targeting MAVS for autophagic degradation [140]. Furthermore, AIV NS1 suppresses IKK-mediated NF-κB activation and interferon-β production by interacting with IKKα and IKKβ [142]. Beyond that, AIV PB2 targets JAK1 for K48-linked ubiquitination and degradation [143]. Remarkably, miR-485 induced by AIV infection targets RIG-I and reduces its expression to dampen RIG-I-mediated antiviral responses [144]. In conclusion, the scope of animal viruses involving antagonistic mechanisms is not limited to the eight types of viruses mentioned above. The summary of these viral antagonistic mechanisms might provide a new theoretical basis for understanding the pathogenic mechanism of animal viruses and developing LAVs based on antagonistic mechanisms and viral virulence factors.

## 9. Concluding Remarks and Future Perspectives

Studies have indicated that animal viruses have evolved various strategies to antagonize host antiviral innate immunity by directly or indirectly blocking the host’s innate immune signaling pathways. The relevant strategies of animal viruses are exhibited in Figure 1, Table 1 and Table 2 [28]. In addition, animal viruses also antagonize it by regulating apoptosis, metabolic pathways, and the formation of stress.

It is worth mentioning that animal viruses can antagonize antiviral innate immunity through mechanisms different from those mentioned above. In addition to regulating transcription, translation, and post-translational modification, viruses could impede the translocation of PRRs. For example, FMDV CD2v interacts with the transmembrane domain of STING and prevents the transport of STING to the Golgi apparatus, thereby inhibiting the cGAS-STING signaling pathway [56]. Furthermore, PRRSV Nsp2 deubiquitinates sensor stromal interaction molecule 1 (STIM1) to inhibit the translocation of STING and interferon-β expression, which ultimately promotes PRRSV replication [145]. Beyond that, there might be more novel antagonistic mechanisms that need to be studied, which lays a solid foundation for developing LAVs in the future.

It is crucial to draw on the research ideas of known antagonistic mechanisms between different viruses based on previous research findings. For instance, compared with other representative viruses, there are relatively few research results for PCV and PPV to antagonize antiviral innate immunity, which highlights the importance of learning from the antagonistic mechanisms of other animal viruses in the future. Remarkably, there are certain similarities between humans and animals in terms of their immune systems. In addition, the mechanism of interaction between hosts and some viruses that seriously affect human health has been extensively studied, which may provide research ideas for animal viruses. Consequently, it is necessary to strengthen the borrowing of research ideas of these viruses while exploring the antagonistic mechanisms of animal viruses [28]. For instance, studies have indicated that IAV infection could weaken the host antiviral innate immunity by hijacking circular RNAs (circRNAs) and vault RNAs (vtRNAs) or antagonizing microRNAs mediating antiviral functions. Specifically, IAV infection may increase circ-MerTK expression to negatively regulate the expression of interferon-β and some ISGs, while vtRNAs induced by IAV infection inhibit PKR activation and PKR-mediated innate immunity [146,147]. Furthermore, several studies have shown that the expression of miR-1307-3p and miR-491 is downregulated after IAV infection, although these microRNAs could directly inhibit viral replication by targeting and reducing viral protein expression [148,149].

Notably, although our understanding of viral antagonistic mechanisms continues to deepen, the potential safety issues of LAVs cannot be ignored. During large-scale vaccination, there may be a risk of virulence reversion of LAVs, which reminds us to conduct sufficient experiments to validate the effectiveness and safety of LAVs before widespread promotion [16]. For example, Ke et al. found that two recombinant viruses, vL126A and vL135A, which are generated through mutations at residues L126 and L135 in PRRSV nsp1β, lead to mild clinical manifestations, low viral titers, a brief period of viremia, and elevated levels of interferon-α and NA titers in the infected pigs. Nevertheless, reversion to the wild-type sequence is detected in certain pigs, and the revertants recover the ability to inhibit interferon production [3,20]. In addition, there are reports that the attenuated vaccine strain rPRVTJ-delgE/gI showed no virulence in pigs, yet it was fatal in sheep [17,150]. These safety issues highlight the key significance of an in-depth exploration of viral antagonistic mechanisms for optimizing the design of LAVs. Only by accurately grasping the viral antagonistic mechanisms and their impact on vaccine characteristics can we have a better chance of effectively avoiding safety hazards and laying a solid foundation for developing safer and more reliable vaccines.

Moving forward, it is essential to constantly extend our knowledge of the battles between host antiviral innate immunity and viral immune antagonism. In this way, when confronted with the next emerging viral outbreak, we could promptly carry out research on the viral antagonistic mechanisms by drawing on previous research ideas, ultimately developing LAVs based on these studied molecular mechanisms. In this way, we may quickly respond to potential threats and the pressure on epidemic prevention and control from unknown emerging viruses in the future.

## Figures and Tables

**Figure 1 vaccines-13-00046-f001:**
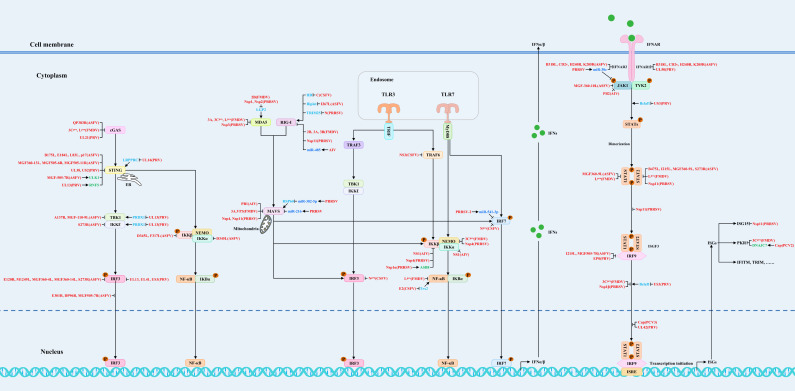
The strategies for animal viruses to block antiviral innate immune signaling pathways. Animal viruses have the potential to inhibit interferon production and interferon responses through regulating various steps of antiviral signaling pathways, including PRRs, signaling proteins, transcription factors, and JAK-STAT signaling pathways. Red, blue, green, and light blue are used to represent viral proteins (viruses), microRNAs, proviral host factors, and host restriction factors, respectively.

**Table 1 vaccines-13-00046-t001:** The strategies for animal viruses to directly block multiple steps of antiviral innate immune signaling pathways.

Steps of Blocked Signaling Pathways	Targets	Viruses (Proteins)	Refs.
PRRs	cGAS	ASFV(QP383R)	[34]
		FMDV(3C^pro^, L^pro^)	[21]
		PRV(UL21)	[100]
	STING	ASFV(B175L, E184L, L83L, MGF360-13L, MGF505-6R, MGF505-11R, and p17)	[35,36,37,38,39,40,41]
		PRV(UL38, US2)	[101,102]
	MDA5	FMDV(3A, 3C^pro^, and L^pro^)	[124,125,126]
		PRRSV(Nsp3)	[75]
	RIG-I	FMDV(2B, 3A, and 3B)	[124,127,128]
		PRRSV(Nsp11)	[76]
Signaling proteins	MAVS	AIV(PB1)	[140]
		FMDV(3A, VP3)	[124,129]
		PRRSV(Nsp4, Nsp11)	[76,77]
	TRAF6	CSFV(NS3)	[115]
	TBK1	ASFV(A137R, MGF-110-9L)	[42,43]
	IKKε	ASFV(S273R)	[44]
	IKKα	AIV(NS1)	[142]
		ASFV(D345L)	[53]
	IKKβ	AIV(NS1)	[142]
		ASFV(D345L, F317L)	[53,54]
		PRRSV(Nsp4)	[78]
	NEMO	FMDV(3C^pro^)	[130]
		PRRSV(Nsp4)	[79]
Transcription factors	IRF3	ASFV(E120R, M1249L, MGF360-4L, MGF360-14L, and S273R)	[45,46,47,48,49]
		CSFV(N^pro^)	[116]
		PRV(UL13, UL41, and US3)	[103,104,105]
	IRF7	CSFV(N^pro^)	[117]
	NF-κB	FMDV(L^pro^)	[22]
Nuclear translocation	IRF3	ASFV(E301R, DP96R, and MGF505-7R)	[50,51,52]
Transcription initiation	Interferon-β promoter	PPV(NS2)	[139]
Interferon responses	IFNAR1	PRV(UL50)	[106]
		ASFV(B318L, CD2v, H240R, and K205R)	[55,56,57,58]
	IFNAR2	ASFV(B318L, CD2v, H240R, and K205R)	[55,56,57,58]
	JAK1	AIV(PB2)	[143]
		ASFV(MGF-360-10L)	[59]
	STAT1	ASFV(MGF360-9L)	[60]
		FMDV(L^pro^)	[23]
	STAT2	ASFV(B475L, I215L, MGF360-9L, S273R)	[60,61,62,63]
		FMDV(L^pro^)	[23]
		PRRSV(Nsp11)	[76]
	ISGF3 formation	PRRSV(Nsp11)	[81]
	IRF9	ASFV(I215L, MGF505-7R)	[64,65]
		PRV(EP0)	[107]
	ISGF3 nuclear translocation	FMDV(3C^pro^)	[131]
		PRRSV(Nsp1β)	[82]
	Binding of ISGF3 to ISRE	PCV3(Cap)	[141]
		PRV(UL42)	[108]
	ISG15	PRRSV(Nsp11)	[83]
	PKR	FMDV(3C^pro^)	[132]
	TRIM25	PRRSV(Nsp1α)	[84]

**Table 2 vaccines-13-00046-t002:** The strategies for animal viruses to indirectly block multiple steps of antiviral innate immune signaling pathways by hijacking ncRNAs and proviral host factors or antagonizing host restriction factors.

Steps of Blocked Signaling Pathways	Targets	ncRNAs or Factors	Viruses (Proteins)	Mechanisms	Ref.
PRRs	STING	ULK1	ASFV(MGF505-7R)	Indirectly blocks multiple steps of antiviral innate immune signaling pathways by hijacking ncRNAs and proviral host factors or antagonizing host restriction factors	[66]
		RNF5	PRV(UL13)		[109]
		LRPPRC	PRV(UL16)		[111]
	MDA5	LGP2	FMDV(2B)		[133]
			PRRSV(Nsp1, Nsp2)		[93]
	RIG-I	miR-485	AIV		[144]
		HB	CSFV(C)		[118]
		Riplet	ASFV(I267L)		[67]
		TRIM25	PRRSV(N)		[94]
Signaling proteins	MAVS	miR-216	PRRSV		[87]
		HSP60	PRRSV		[86]
	IKKβ	ASB8	PRRSV(Nsp1α)		[85]
	IKKε	PRDX1	PRV(UL13)		[112]
	TBK1	PRDX1	PRV(UL13)		[112]
Transcription factors	IRF7	miR-541-3p	PRRSV-2		[88]
Interferon responses	NF-κB	Trx2	CSFV(E2)		[119]
	IFNAR2	miR-30c	PRRSV		[89]
	ISGF3 nuclear translocation	Bclaf1	PRV(US3)		[113]
	PKR	DNAJC7	PCV2(Cap)		[138]

## Data Availability

Data sharing is not applicable to this article as no new data were created or analysed in this study.

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
