# Peer review of "The Strategies Used by Animal Viruses to Antagonize Host Antiviral Innate Immunity: New Clues for Developing Live Attenuated Vaccines (LAVs)"

_vaccines, 2025, doi:10.3390/vaccines13010046_

Round 1
Reviewer 1 Report
Comments and Suggestions for Authors
This article is a short Review on how some animal viruses can antagonize host antiviral responses and how this knowledge can help developing live attenuated vaccines.
The merit of the Review is that it appropriately reviews the former, with the major viruses of veterinarian relevance, ASFV, PPRSV, PRV, CSFV, and FMDV taken as example. However, the knowledges on some other viruses, i.e., PCV, PPV, AIV, ALV, and IBDV, are too briefly mentioned to convey a truly interesting and relevant information. Also, I found that the latter intention, i.e., how this knowledge could help developing live attenuated vaccines is not or very poorly explained though the authors state that while much more is known for human viruses, this could “provide research ideas for animal viruses” (line 479). While the authors infer that “there are certain similarities between humans and animals in terms of their immune systems” (lines 476-477), I believe that this is a truly naïve vision of the field. Also, the authors should take into account that most, if not all, of the so-called human viral pathogens cannot seriously be referred as “human viruses”, as they arise via zoonoses. In this way, the authors should revise these statements but also, and interestingly enough for this Review, discuss how could transmission between species impact the sensitivity to antagonize host antiviral responses and how differential susceptibilities and/or antagonisms to the above could shape up patterns of transmission and pathogenesis.
Minor points: several terms are not defined and the authors should revise their manuscript so that abbreviations are well introduced when appropriate in the text (e.g., ALV, vtRNA, etc).
Comments on the Quality of English LanguageThis needs attention at some places
Reviewer 2 Report
Comments and Suggestions for Authors
The review article by Chen and Zhang seeks to establish the importance of innate immune evasion effectors as targets for the development of Live-attenuated vaccines (LAVs). On the whole the review is very dense, and while largely complete and exhaustive for the viruses listed, limited in scope as relatively few animal viruses are described in depth. It appears to be more focused on diseases of livestock versus zoonoses at large.
A critical element missing from the review is a proof-of-concept example demonstrating that immune evasion genes are indeed excellent targets for the development of novel live attenuated vaccines. On face value reducing the capacity to evade innate immunity is easily linked to LAV development, but no specific examples outlining where this has been explicitly demonstrated are provided. Questions that should be addressed include- What strategies do current LAVs use, do any follow the suggested paradigm, or is the mechanism(s) of attenuation due to differences elsewhere?
Similarly, does altering the capacity of the pathogen to evade immunity defeat the robustness, efficacy, or longevity of the immune response? Examples supporting (or contradicting) the stated purpose of the review exist but are notable absent.
The arguments made by the article are very straightforward, but in the absence of clear directed evidence in support of the claims (or a critical assessment as to what must be considered) the utility of the review is a bit stunted.
Minor comment- Virus names and other abbreviations should be clearly spelled out the first time that they are used and not at a table in the end.
Comments on the Quality of English LanguageEditing for grammar and organization would improve the readability.
Reviewer 3 Report
Comments and Suggestions for Authors
A review manuscript (vaccines-334115) entitled “The strategies for animal viruses to antagonize the host antiviral innate immunity: an indispensable research direction for developing live attenuated vaccines” by N. Chen and B. Zhang describes gives the light to the innate immunity and the host restriction factors for animal viruses and try to contribute to the research on the efficacy and the safety of live attenuated animal vaccines. The authors collected many published results on the strategies for how viruses break through the host defense and tried to summarize them in relation to the effective live attenuated vaccines. The approach is interesting and worth publishing in the journal, Vaccines.
However, before accepting for publication, this reviewer would like to raise a few major and minor issues that need clarification.
Major points;
1. Authors try to review the animal virus diseases. However, the description is only focused on the swine pathogens (ASFV, PRRSV, PRV, CSFV, FMDV, PCV2, PCV3, and PPV) and the avian pathogens (AIV, ALV, and IBDV). The manuscript does not mention other domestic animals (like bovine and equine), companion animals (dogs and cats), fishes, and wild animals. The animals that the authors focused on should be shown clearly in the Introduction section.
2. The authors subtitle this manuscript as “an indispensable research direction for developing live attenuated vaccines”, but the research direction of how to improve the present live attenuated vaccines or how to develop the future live attenuated vaccines is not clearly shown in each sub-section. The present style of this review article is just the summary of some swine and avian virus antagonistic mechanisms against the host innate immunity reaction. The indispensability of the research direction for the anti-host innate immunity of viruses should be clarified in detail.
3. The authors do not introduce the way how classical live attenuated vaccines have developed. The virulence of viruses was attenuated classically by passing many times in different hosts or tissues or at various temperatures. Different host or tissue-adapted viruses show the attenuated phenotype for the natural host infection. Different temperature-adapted viruses show the attenuated phenotype during the natural infection, thereby showing the ts (temperature sensitive)-phenotype. For example, the human live vaccines, measles, mumps, and rubella vaccines have been established by adapting other hosts (i.e., chick embryonic fibroblasts, rabbit kidney) and successfully used worldwide. Live influenza vaccine (FluMist) with ts-phenotype can grow only in the upper respiratory tissue with the low temperature but not in the lower respiratory tissues with average body temperature and induce protective immunity safely. These classical strategies do not relate to the host innate immunity. These should be explained as background information.
Minor comments:
1. Line 34; The word “animal diseases” range is broad, so it is better to narrow it as “an infectious livestock disease”.
2. Lines 74, 79, and 78; AIV is listed in the Glossary as avian influenza virus, but IAV is not. Please add the IAV to the glossary section. Otherwise, please spell it like influenza A virus (IAV).
3. Line 80; The words “wild-type virus challenges” had better be described as “wild-type IAV challenges”.
4. Lines 103-107, Figure 1; An endosome with TLR3/7 is drawn in the cytoplasm, but STING and MAVS are not drawn with ER and mitochondria, respectively. A type 1 interferon receptor, IFNAR, is drawn as a single molecule but is described in the text as a hetero dimer of IFNAR1 and IFNAR2. Please re-draw the IFNAR as the dimer of IFNAR1 and IFNAR2. IRF3 and NRF-7 are drawn to be phosphorylated in the cytoplasm and translocated into the nucleus. However, IRF3 and NRF-7 attached to the gene are no longer phosphorylated. Do dephosphorylation of IRF3 and NRF-7 occur?
5. Lines 115-117; Please add the DNA genome size of ASFV
6. Lines 159 and others: A term, “interferon” and “IFN”, is used together throughout the text even though the two terms mean the same. Please unit either. The use of INF-alpha/beta and type 1 interferon is confusing.
7. Lines 163-164; A sentence “the recruitment of TYK2 and JAK1 by IFNAR1 and IFNAR2 by interacting with IFNAR1 and IFNAR” should be clarified.
8. Lines 210-211; The Jack2-STAT3 pathway is shown without explanation. Please explain briefly about this pathway. No Jack2-STAT3 pathway is shown in Figure 1.
9. Lines 218-221; Please add the RNA genome size of PRRSV.
10. Lines 234-235; Nsp11 was shown in ref #58 to reduce MAVS and RIG-I expression, but not shown in ref #57. Please check the citation.
11. Lines 293-295; Please add the DNA genome size of PRV.
12. Line 345; Please make a new sub-section describing the brief introduction of CSFV and show the RNA genome size of CSFV and the present situation of live attenuated CSFV vaccines.
13. Line 372; Please make a new sub-section describing the brief introduction of FMDV and show the RNA genome size of FMDV and the present situation of live attenuated FMDV vaccines.
14. Line 416; Swine viruses, PCV and PPV, and avian viruses IAV, ALV, and IBDV” are collectively explained in section 8. If this section is essential, please separate reasonably. This reviewer knows that ALV was contaminated with live attenuated vaccines such as the Fowlpox virus vaccine, Newcastle disease vaccine, and IBDV vaccine, but does not know the development of the ALV vaccine. Please show the author's view about the live attenuated ALV vaccine.
15. Lines 446-451 Figure 2; An endosome with TLR3/7 is drawn in the cytoplasm, but STING and MAVS are not drawn with ER and mitochondria, respectively. IRF3 and NRF-7 are drawn to be phosphorylated in the cytoplasm and translocated into the nucleus. However, IRF3 and NRF-7 attached to the gene are no longer phosphorylated. Do dephosphorylation of IRF3 and NRF-7 occur?
16. Lines 452-456 Figure 3; A type 1 interferon receptor, IFNAR, is drawn as a single molecule but is described in the text as a hetero dimer of IFNAR1 and IFNAR2. Please re-draw the IFNAR as the dimer of IFNAR1 and IFNAR2. Is IFN-alpha/beta induction shown on the left side of Figure 3 necessary? This figure should concentrate on the Jak/STAT signaling pathway and ISG gene induction.
17. Pages 11-13, Tables 1; Please show the meaning of N/A in the center of page 12.
18. Pages 14-15, Glossary; How did the authors determine the order of the abbreviations list? It is not easy for the reader to find the spell-out form. Please consider lining them in alphabetical order.
19. Please show the references with DOI.
Round 2
Reviewer 3 Report
Comments and Suggestions for Authors
This reviewer confirmed that the authors properly corrected the review manuscript (vaccines-334115) entitled “The strategies for animal viruses to antagonize the host antiviral innate immunity: New clues for developing live attenuated vaccines (LAVs)” according to the suggestion and comments raised by the reviewers.
The revision clarified the story of the review article. The authors succeed in collecting many published results and explaining them with the schematic diagrams.
This reviewer thinks that the article will contribute to developing effective live attenuated vaccines for infectious livestock diseases and would like to suggest “Accept in present form” for the journal, Vaccines.
